# Nasopharyngeal Expression of Angiotensin-Converting Enzyme 2 and Transmembrane Serine Protease 2 in Children within SARS-CoV-2-Infected Family Clusters

Mohammad Rubayet Hasan,[a,b] Muneera Naseer Ahmad,[a,c] Soha Roger Dargham,[b] Hatem Zayed,[c] Alaa Al Hashemi,[a] Nonhlanhla Ngwabi,[a] Andres Perez Lopez,[a,b] Simon Dobson,[a] Laith Jamal Abu Raddad,[b] Patrick Tang[a,b]

[a]Sidra Medicine, Doha, Qatar
[b]Weill Cornell Medicine-Qatar, Doha, Qatar
[c]Department of Biomedical Sciences, College of Health Sciences, Qatar University, Doha, Qatar

**ABSTRACT** Lower levels of angiotensin-converting enzyme 2 (ACE2) and transmembrane serine protease 2 (TMPRSS2) in the nasal epithelium of children may be related to a lower incidence of severe acute respiratory syndrome coronavirus 2 (SARS-CoV-2) infection, compared to adults. However, no direct evidence is available to support this hypothesis. In this study, we compared the transcript levels of ACE2 and TMPRSS2 in nasopharyngeal swab samples ($n = 234$) from children and adult family members within SARS-CoV-2-exposed families and assessed the association with SARS-CoV-2 infection status. Transcript levels for ACE2, but not TMPRSS2, were higher in adults than in children ($n = 129$ adults and 105 children; $P < 0.05$). The expression of the two genes was not significantly different between SARS-CoV-2 positive and SARS-CoV-2 negative patients within the same age groups. However, in families with one or more SARS-CoV-2 positive adult family members, expression of both genes was significantly higher in SARS-CoV-2 positive children than in SARS-CoV-2 negative children ($P < 0.05$). By multivariate analysis, ACE2 expression adjusted for age and sex was significantly associated with SARS-CoV-2 infection in the overall population (odds ratio [OR], 1.112 [95% confidence interval [CI], 1.012 to 1.229]; $P < 0.05$). The degree of this association was higher (OR, 1.172 [95% CI, 1.034 to 1.347]; $P < 0.05$) in the subgroup of families with only SARS-CoV-2 positive adult family members. Our results suggest that children with lower levels of nasal ACE2 and TMPRSS2 are more likely to remain SARS-CoV-2 negative despite being exposed to a SARS-CoV-2 positive adult family member.

**IMPORTANCE** ACE2 and TMPRSS2 are well established in the literature as SARS-CoV-2 entry factors. Recent data suggest that lower levels of nasal ACE2 in children may be associated with their lower incidence of coronavirus disease 2019 (COVID-19). In this study, using data from nasopharyngeal swab specimens from adult and pediatric members of families in which one or more members of the family had laboratory-confirmed SARS-CoV-2 infection, we show that children with lower levels of ACE2 and TMPRSS2 are more likely to remain SARS-CoV-2 negative despite being exposed to a SARS-CoV-2 positive adult family member. These results provide new insights into the roles of nasopharyngeal ACE2 and TMPRSS2 in acquiring SARS-CoV-2 infection, and they show that the differential expression of these genes in adults versus children may contribute to differential rates of SARS-CoV-2 infection in these populations.

**KEYWORDS** angiotensin-converting enzyme 2, COVID-19, SARS-CoV-2, transmembrane serine protease 2, nasopharyngeal

Address correspondence to Mohammad Rubayet Hasan, mhasan@sidra.org.

The global pandemic of coronavirus disease 2019 (COVID-19) (1) caused by the novel severe acute respiratory syndrome coronavirus 2 (SARS-CoV-2) prompted urgent research on the pathogenesis and transmission of the virus, including

elucidation of the mechanisms by which SARS-CoV-2 binds to and enters host cells. SARS-CoV-2 is genetically and structurally related to SARS-CoV, and both share the same cell surface receptor, angiotensin-converting enzyme 2 (ACE2), for binding and entry into the host cells through the spike (S) glycoprotein (2–4). However, mutations in the receptor binding domain of the S gene that provide higher binding affinity for ACE2 have led to increased infectivity and enhanced transmissibility of SARS-CoV-2. The SARS-CoV-2 spike also exhibits a unique furin cleavage site that is proteolytically cleaved by a transmembrane protease serine 2 (TMPRSS2), which may enhance the human-to-human transmission (3, 4). The expression of both ACE2 and TMPRSS2 has been reported in various human organs, including the lungs, heart, kidneys, ileum, and bladder (5). Within the respiratory tract, based on both single-cell RNA sequencing and immunohistochemistry, ACE2 expression was shown to be highest in the sinonasal epithelium and alveolar type II cells. TMPRSS2 is also colocalized with ACE2 on the apical surface of a small subset of alveolar type II cells in the lung parenchyma, suggesting a role in regulating disease severity (6, 7). SARS-CoV-2 infects many different cell types. Apart from the TMPRSS2-ACE2 pathway, multiple mechanisms exist for SARS-CoV-2 viral entry into cells, depending on the cell type (8).

The clinical manifestations of COVID-19 are heterogeneous, from asymptomatic to moderate to highly severe, life-threatening disease. Although individuals of any age can acquire SARS-CoV-2 infection, most cases of COVID-19 have been reported in adults and elderly people (9). The incidence of COVID-19 is much lower in children (10 to 13% of laboratory-confirmed cases in children <18 years of age) than in adults (10). Also, the rates of the severe forms of COVID-19 and mortality rates associated with SARS-CoV-2 infection are much lower among children than among adults. In most cases, children remain asymptomatic or mildly or moderately symptomatic (11). The reason for the differences in the rates of infection and severity among adult versus pediatric patients is not well known. However, a number of hypotheses exist, including differences in the immune response and cytokine regulation between adults and children (12), the presence of preexisting cross-reactive antibodies (13), and off-target vaccine-derived immunity in children (14). The role of ACE2 receptors expressed in the airways and lungs has also been examined (15). A recent study of ACE2 gene expression in the nasal epithelium of children and adults, using samples from a pre-COVID-19 asthma cohort, showed that ACE2 expression in the nasal epithelium correlates with age, suggesting that the lower expression of the gene in the nasal epithelium of children may explain why children are less affected by COVID-19 (15). Similarly, in cultured cells, it was shown that TMPRSS2-expressing cell lines are highly susceptible to SARS-CoV-2 infection (16). Using a public gene expression data set, it was also shown that the nasal and bronchial expression of ACE2 and TMPRSS2 is lower in children than in adults (17). Considering the respiratory and olfactory epithelium in the nasal mucosa as one of the first contact points for the virus, the expression of these genes in nasal epithelium may have an important role in SARS-CoV-2 infection and transmission. Direct evidence that correlates nasal ACE2 and TMPRSS2 expression with SARS-CoV-2 infection or symptoms associated with COVID-19 is limited. A recent study demonstrated that the mean transcript levels of nasopharyngeal ACE2 and TMPRSS2 are lower in SARS-CoV-2 positive patients, but transcription of the transmembrane form of ACE2 positively correlates with SARS-CoV-2 viral load (18). Despite these data, it remains unclear whether lower postexposure SARS-CoV-2 infection rates are related to lower nasal expression of ACE2 and TMPRSS2 in children. Therefore, in this study, we analyzed the transcript levels of ACE2 and TMPRSS2 in nasopharyngeal swab (NPS) specimens from SARS-CoV-2 positive and negative adults and children and assessed their association with SARS-CoV-2 infection status within these family clusters.

## RESULTS

**Patient demographic and clinical characteristics.** NPS specimens were collected from children and adults from families with a history of laboratory-confirmed SARS-

**TABLE 1** Participant characteristics

| Parameter | Data for: | |
| --- | --- | --- |
| | Adults | Children |
| No. of samples | 129 | 105 |
| Sex (no. [%]) | | |
| Female | 81 (62.8) | 49 (46.7) |
| Male | 48 (37.2) | 56 (53.3) |
| Age (median [IQR]) (yr) | 34 (30–39) | 4 (1.25–8) |
| SARS-CoV-2 PCR result (no. [%]) | | |
| Negative | 45 (34.9) | 49 (46.7) |
| Positive | 84 (65.1) | 56 (53.3) |

CoV-2 infection in at least one of the family members. A total of 234 NPS specimens were collected, of which 129 were from adults and 105 were from children, from a total of 105 families (Table 1). NPS specimens from more than one adult family member were available from 24 families. Among the study subjects, 62.6% and 46.7% were female in the adult and pediatric groups, respectively. The median ages of the adults and children were 34 years (interquartile range [IQR], 30 to 39 years) and 4 years (IQR, 1.25 to 8 years), respectively; 65.1% of adults and 53.3% of children were SARS-CoV-2 positive by reverse transcription quantitative PCR (RT-qPCR). The median ages of SARS-CoV-2 positive and negative adults were 34 years (IQR, 29.5 to 40 years) and 33 years (IQR, 30.25 to 38 years), respectively, and were not significantly different from each other. The median ages of SARS-CoV-2 positive and negative children were 5 years (IQR, 1.25 to 8 years) and 3.5 years (IQR, 1.12 to 7.755 years), respectively, and were not significantly different from each other.

All patients and visitors presenting to Sidra Medicine were triaged for a history of COVID-19 and COVID-19-associated symptoms, including temperature assessment. As a designated COVID-19-free facility, only patients with no history of COVID-19 within the past 2 weeks and patients with no COVID-19-related symptoms are eligible for admission to the hospital. Therefore, the children and adults included in this study were asymptomatic at the time of NPS specimen collection. No other clinical data on COVID-19-associated symptoms were collected.

**Relative gene expression of ACE2 and TMPRSS2 in NPS specimens.** For gene expression analysis of ACE2, the transmembrane isoform of ACE2 was specifically targeted because the soluble form of the protein may have a contrasting role in SARS-CoV-2 infection and no role in viral entry into the cells (19, 20). The chosen ACE2 target is located at the exon 17-exon 18 boundary (GenBank accession number NM_021804.2) within the transmembrane domain of ACE2 (Hs01085333_m1; Thermo Fisher Scientific). When the assay was applied to 234 specimens, nasopharyngeal expression of $\beta$-actin was detected in all specimens, with a median threshold cycle ($C_T$) value of 25.9 (IQR, 24.4 to 27.9). However, the RT-qPCR $C_T$ values for ACE2 and TMPRSS2 remained undetermined in 135 and 43 samples, respectively. Therefore, for quantitative analysis, a $C_T$ value of 40 was assigned to data representing these specimens. Based on RT-qPCR $C_T$ values, the relative transcript levels of ACE2 and TMPRSS2 were lower and more widely variable than those of $\beta$-actin (Fig. 1A). The $C_T$ values of $\beta$-actin were normally distributed across all specimens (Shapiro-Wilk test, W = 0.9949 [P = 0.6243]; Anderson-Darling test, A2 = 3,786 [P = 0.4039]). The mean $C_T$ values for $\beta$-actin were not significantly different between SARS-CoV-2 positive and negative patients (P = 0.2967, unpaired 2-tailed t test). However, the mean $\beta$-actin $C_T$ values (25.3 $\pm$ 2.8) for NPS specimens from children were significantly stronger than the $\beta$-actin $C_T$ values (26.8 $\pm$ 2.5) for specimens from adults (P < 0.0001, unpaired 2-tailed t test) (see Fig. S1 in the supplemental material).

Although the $C_T$ values for $\beta$-actin were normally distributed, $\Delta C_T$ for ACE2 and TMPRSS2 failed the normality test (W = 0.9822 [P < 0.0049] for ACE2 and W = 0.9807

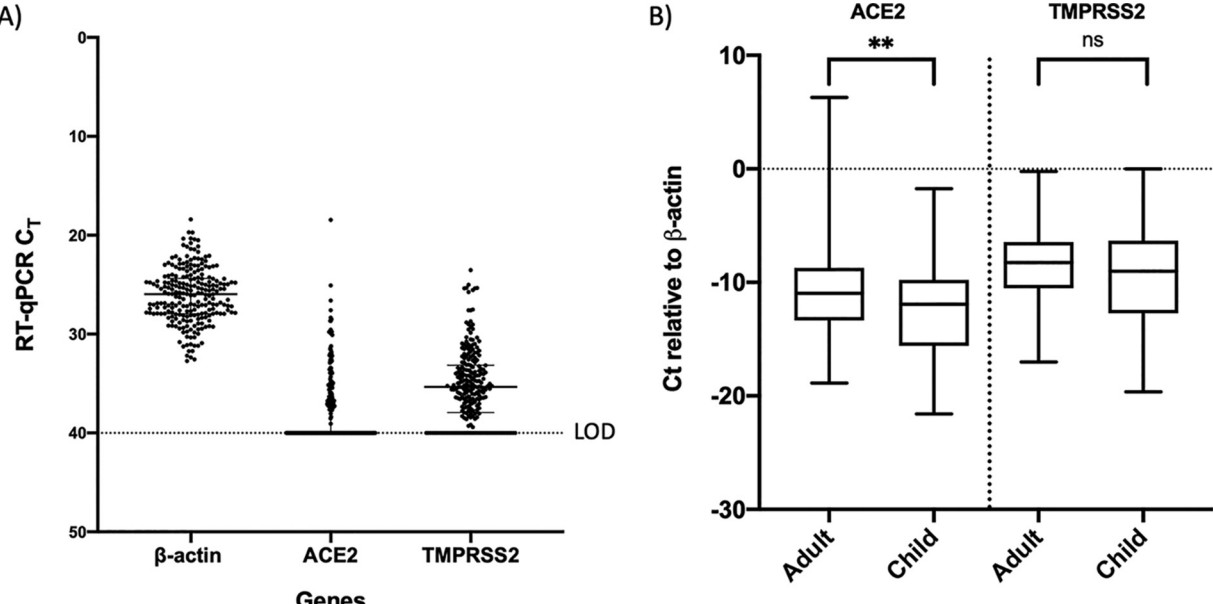

**FIG 1** Gene expression of ACE2 and TMPRSS2 in children and adults within family clusters exposed to SARS-CoV-2. (A) Variations in the transcript levels of $\beta$-actin, ACE2, and TMPRSS2 in NPS specimens in the study population. Data show median $C_T$ values with IQR ($n = 234$). (B) Comparison of transcript levels of ACE2 and TMPRSS2 in NPS specimens, relative to $\beta$-actin ($\Delta C_T$), for adults and children. $\Delta C_T$ values were calculated by subtracting the $C_T$ values for ACE2 or TMPRSS2 from the respective $C_T$ values for the housekeeping gene $\beta$-actin. Data show median $C_T$ values with IQR ($n = 129$ adults and 105 children). $P$ values were calculated with the Mann-Whitney $U$ test. **, $P \leq 0.001$; ns, not significant.

[$P = 0.0028$] for TMPRSS2) (see Fig. S2). Therefore, a nonparametric test i.e., the Kruskal-Wallis test with Dunn's multiple-comparisons test or the Mann-Whitney $U$ test, was used to compare multiple groups or two groups, respectively. When the transcript levels of ACE2 and TMPRSS2 were compared for the overall study population, only ACE2 transcript levels were significantly higher ($P < 0.0049$) in the adult population (Fig. 1B). ACE2 transcript levels were also significantly higher ($P < 0.05$) in the SARS-CoV-2 positive population irrespective of age. However, expression of neither gene was significantly different between SARS-CoV-2 positive and SARS-CoV-2 negative patients within the same age group (see Table S1). The median values of ACE2 but not TMPRSS2, transcript levels were significantly different between SARS-CoV-2 positive and SARS-CoV-2 negative adults and children by Kruskal-Wallis one-way analysis of variance (ANOVA) ($P < 0.01$); by Dunn's multiple-comparison test, a significant difference was seen only for ACE2 transcript levels between SARS-CoV-2 positive adults and SARS-CoV-2 negative children ($P < 0.01$) (Fig. 2A and B). Similar results were obtained with the Mann-Whitney $U$ test when the transcript levels were compared between SARS-CoV-2 positive adults and SARS-CoV-2 positive children, between SARS-CoV-2 positive adults and SARS-CoV-2 negative children, or between SARS-CoV-2 negative adults and SARS-CoV-2 positive children (data not shown). Only ACE2 expression was significantly higher in SARS-CoV-2 positive adults than in SARS-CoV-2 negative children ($P < 0.05$). Interestingly, when SARS-CoV-2 positive and negative children from families with one or more adults positive for SARS-CoV-2 were compared, the transcript levels of both genes were significantly lower in the SARS-CoV-2 negative children ($P < 0.05$) (Fig. 2C). To determine whether age differences among the SARS-CoV-2 positive and negative children were responsible for such a difference, we compared the age distribution of the subjects in these groups. The median ages of SARS-CoV-2 positive children and SARS-CoV-2 negative children from these families were 3 years (IQR, 1 to 6 years) and 5 years (IQR, 1.25 to 8 years), respectively, and were not significantly different from each other. We also assessed whether the association between ACE2 and SARS-CoV-2 infection was affected by the viral load in the adults who were positive for SARS-CoV-2. Of

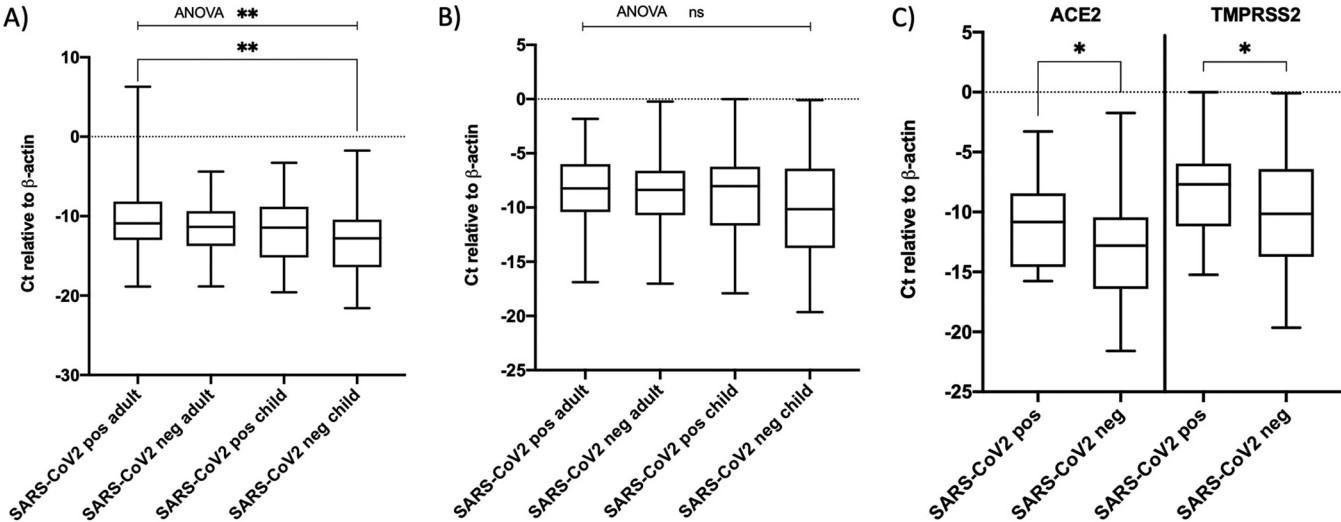

**FIG 2** Comparison of transcript levels of ACE2 and TMPRSS2 between SARS-CoV-2 positive and negative adults and children within family clusters exposed to SARS-CoV-2. (A) Comparison of ACE2 transcript levels in SARS-CoV-2 positive adults, SARS-CoV-2 negative adults, SARS-CoV-2 positive children, and SARS-CoV-2 negative children. (B) Comparison of TMPRSS2 transcript levels in SARS-CoV-2 positive adults, SARS-CoV-2 negative adults, SARS-CoV-2 positive children, and SARS-CoV-2 negative children (nonparametric ANOVA Kruskal-Wallis test followed by Dunn's multiple-comparison test). (C) Comparison of ACE2 and TMPRSS2 transcript levels in SARS-CoV-2 positive and negative children from families with one or more adults positive for SARS-CoV-2 (Mann-Whitney $U$ test). *, $P \leq 0.05$; **, $P \leq 0.001$; ns, not significant.

84 SARS-CoV-2 positive adults, SARS-CoV-2 RT-qPCR $C_T$ values were available for 68 individuals and were used as surrogate markers for viral load. We compared the SARS-CoV-2 RT-qPCR $C_T$ values for adults from families with SARS-CoV-2 negative and positive children. However, neither the means nor the medians of the $C_T$ values were significantly different between these two groups (see Fig. S3).

**Association of nasopharyngeal ACE2 and TMPRSS2 expression with SARS-CoV-2 infection status.** By multiple logistic regression analysis with SARS-CoV-2 infection as the outcome variable and age, sex, and ACE2 or TMPRSS2 transcript levels as independent variables, ACE2 transcript levels were independently associated with SARS-CoV-2 positivity, with an adjusted odds ratio (OR) of 1.094 (95% confidence interval [CI], 1.02 to 1.177; $P = 0.0142$) (Table 2). However, TMPRSS2 expression was not associated with SARS-CoV-2 positivity. In a subpopulation of families with at least one adult positive for SARS-CoV-2, through multiple logistic regression analysis with the SARS-CoV-2 result as the outcome variable and sex and transcript levels of ACE2 or TMPRSS2 as dependent variables, the OR was higher (OR, 1.19 [95% CI, 1.082 to 1.322]; $P = 0.0006$) than in the overall population. In this subpopulation, TMPRSS2 expression was also significantly associated with SARS-CoV-2 positivity (OR, 1.13 [95% CI, 1.035 to 1.238]; $P = 0.0073$) (Table 2). For further confirmation of these results, we excluded samples that failed to show detectable ACE2 and TMPRSS2 by gene expression analysis. Despite a significant reduction in sample numbers, an association between ACE2 transcript levels and SARS-CoV-2 infection was still noted, particularly in the subpopulation of families with at least one adult positive for SARS-CoV-2 (adjusted OR, 1.159 [95% CI, 1.014 to 1.351]; $P = 0.0419$) (see Table S3).

Apart from SARS-CoV-2 infection status, the expression of both ACE2 and TMPRSS2 was significantly associated (adjusted $\beta$-coefficient, 1.495 [95% CI, 0.4575 to 2.533] [$P = 0.0049$] and 1.038 [95% CI, 0.02043 to 2.055] [$P = 0.0456$], respectively) with the adult population using multiple linear regression analyses with ACE2 or TMPRSS2 transcript levels as the outcome variables and age group and sex as independent variables, consistent with the gene expression data (Fig. 1B). However, no significant association of ACE2 and TMPRSS2 expression with sex was found. We also queried whether the combination of ACE2 and TMPRSS2 transcript levels was related to COVID-19 positivity, using a two-way interaction multivariate regression model, but no significant association was observed.

**TABLE 2** Association of NPS specimen ACE2 and TMPRSS2 gene expression with SARS-CoV-2 infection status

| Group and variable | Overall (*n* = 234) | | | Subgroup of families with ≥1 adult positive for SARS-CoV-2 (*n* = 172) | |
|---|---|---|---|---|---|
| | OR^*a* (95% CI) | *P* | | OR^*a* (95% CI) | *P* |
| ACE2 | | | | | |
| Age | 1.009 (0.9919–1.026) | 0.3094 | | | |
| Sex (male) | 1.091 (0.6358–1.883) | 0.7517 | | 0.6817 (0.3349–1.383) | 0.2879 |
| ACE2 gene expression | 1.094 (1.02–1.177) | 0.0142 | | 1.19 (1.082–1.322) | 0.0006 |
| TMPRSS2 | | | | | |
| Age | 1.012 (0.9951–1.029) | 0.1722 | | | |
| Sex (male) | 1.043 (0.6114–1.785) | 0.8779 | | 0.6469 (0.3217–1.296) | 0.2188 |
| TMPRSS2 gene expression | 1.046 (0.9775–1.121) | 0.1945 | | 1.13 (1.035–1.238) | 0.0073 |

^*a*ORs were calculated from multiple logistic regression analysis.

In addition to SARS-CoV-2 infection status, we have assessed whether ACE2 or TMPRSS2 transcript levels were associated with viral load in the SARS-CoV-2 positive individuals for whom RT-qPCR $C_T$ values were available (*n* = 112). By multiple linear regression analysis, with SARS-CoV-2 RT-qPCR $C_T$ values as the outcome variable and age, sex, and ACE2 or TMPRSS2 transcript levels as independent variables, ACE2 expression but not TMPRSS2 expression was significantly negatively associated with the SARS-CoV-2 $C_T$ values (adjusted $\beta$-coefficient, −0.5069 [95% CI, −0.8438 to −0.1699] [*P* = 0.0036] for ACE2 and −0.1929 [95% CI, −0.5481 to 0.1623] [*P* = 0.2839] for TMPRSS2).

## DISCUSSION

Although the role of ACE2 in SARS-CoV-2 infection was hypothesized and investigated soon after the declaration of the COVID-19 pandemic (2, 3), little is known about the role of the protein in the nasopharyngeal epithelium. Considering the respiratory and olfactory epithelium in the nasal mucosa as one of the first contact points for the virus, ACE2 expression in nasal epithelium may have an important role in SARS-CoV-2 infection and transmission. We conducted an observational, case-control study to investigate whether lower nasopharyngeal ACE2 and TMPRSS2 levels in children protect them from acquiring SARS-CoV-2 infection despite being exposed to SARS-CoV-2 positive adult family members. We included families in which family members were differentially affected by COVID-19, so that SARS-CoV-2 negative family members in this setting serve as case controls, as these individuals remained SARS-CoV-2 negative despite being exposed to the virus. It should be noted that none of the study subjects was ≥65 years of age, and it is very unlikely that any of the study subjects were vaccinated at the time of NPS sample collection, because the samples were acquired prior to the availability of the COVID-19 vaccines in Qatar.

For gene expression analysis of ACE2, the transmembrane isoform of ACE2 was specifically targeted because the soluble form of the protein might have a contrasting role in SARS-CoV-2 infection and no role in viral entry into host cells (18–20). Consistent with earlier reports, nasopharyngeal ACE2 expression in our study was higher in adults than in children (15, 17). In addition, our study provides direct evidence that the nasopharyngeal expression of TMPRSS2 is higher in adults than in children. In our study population, ACE2 transcript levels were significantly higher in SARS-CoV-2 positive patients and ACE2 transcript levels were significantly associated with SARS-CoV-2 positivity. These results are in contrast to the findings from an independent study conducted in British Columbia, Canada, which showed that nasopharyngeal ACE2 expression is lower in COVID-19 patients (18). This may be explained by the fact that the British Columbia study was conducted in a predominantly adult population (mean age, ~62 years), while both adults and children from preselected COVID-19 family clusters were studied in our study. Furthermore, SARS-CoV-2 negative participants in our study represent a specific group of people who remained SARS-CoV-2 negative despite being

likely exposed to the virus in their families, as opposed to the randomly selected SARS-CoV-2 negative people in the British Columbia study.

By ANOVA and multiple-comparison tests, we demonstrated that ACE2 expression was lowest in the SARS-CoV-2 negative pediatric group, compared to SARS-CoV-2 positive children and SARS-CoV-2 positive or negative adults. A similar pattern was seen for TMPRSS2 as well, but the difference in TMPRSS2 expression in the SARS-CoV-2 negative pediatric group, compared to the other groups, was not statistically significant. Perhaps the most important observation of our study was the significantly lower expression of both ACE2 and TMPRSS2 in the SARS-CoV-2 negative children, compared to SARS-CoV-2 positive children for whom one or more adult family members were positive for SARS-CoV-2 (Fig. 2C). Also, by multivariate analysis for this subpopulation, higher expression of both genes was significantly associated with SARS-CoV-2 positive children. The OR for ACE2 expression was higher in this subpopulation than the overall population (Table 2). These results suggest that children with lower expression of ACE2 and TMPRSS2 may remain SARS-CoV-2 negative despite being exposed to the virus in their families. The effects were subtle but statistically significant, suggesting that other associated factors, such as differences in immunological responses, may also be involved. The viral load in SARS-CoV-2 positive adults could be a confounding factor as well, because of its role in viral transmission within the exposed families. However, we ruled out this possibility by comparing SARS-CoV-2 RT-qPCR $C_T$ values (as a proxy for viral load) between the SARS-CoV-2 positive adults from families with SARS-CoV-2 positive versus negative children. Interestingly, however, we noted a statistically significant negative association between ACE2 gene expression and higher SARS-CoV-2 RT-qPCR $C_T$ values (lower viral loads), providing further support for our observations.

Our study has several limitations. First, the expression of ACE2 and TMPRSS2 remained undetermined in a significant proportion of NPS specimens. Because the expression of $\beta$-actin in those specimens (median $C_T$, 26.4 [IQR, 24.4 to 28.5]) was not different from that in the rest of the samples, ACE2 and TMPRSS2 RT-qPCR assay results in those samples represent lower expression of those genes rather than RNA degradation. Although the calculation of relative transcript levels of ACE2 and TMPRSS2 in those specimens may be less accurate, approximate values can still be inferred because of the variation in the expression of the $\beta$-actin gene. Another limitation of the study is that the SARS-CoV-2 positive patients in our setting were mostly asymptomatic, and the study was conducted in a relatively small population. Also, clinical data on disease state and exposure level were not available, and the laboratory results for the subjects who were initially negative for SARS-CoV-2 could not be prospectively studied. Future studies assessing the roles of nasopharyngeal ACE2 and TMPRSS2 in SARS-CoV-2 infection and COVID-19 disease severity should include a larger number of random family clusters that include symptomatic patients, as well as those with different levels of severity.

In conclusion, our results support the previously described hypothesis that children have lower nasopharyngeal ACE2 and TMPRSS2 transcript levels, which may protect them against infection from SARS-CoV-2 when they are exposed to SARS-CoV-2 positive adult family members. These results may also provide indications for future studies on the role of these proteins as prognostic markers for the disease.

## MATERIALS AND METHODS

**Study design and collection of specimens.** Sidra Medicine is a 400-bed, pediatric tertiary care hospital in Qatar. In 2020, as part of an integrated national pandemic management plan, the hospital was designated a COVID-19-free facility. Since 16 April 2020, all patients and accompanying adult family members visiting the hospital have been actively screened for SARS-CoV-2 by RT-qPCR testing of NPS specimens. During the period from June to December 2020, residual NPS specimens that met the inclusion criteria of the study were deidentified and stored at −80°C for further analysis. The inclusion criteria were (i) at least one member of the family was positive for SARS-CoV-2 by RT-qPCR, (ii) paired NPS specimens were available for the child and at least one of the accompanying adult family members, and (iii) at least 0.5 ml of specimen was available for gene expression analysis. Paired specimens from SARS-CoV-2 positive and negative children and their adult companions were identified from the hospital infection prevention and control records. Data on the age, sex, and SARS-CoV-2 RT-qPCR results for the selected

patients were extracted from the laboratory information system. Ethics approval for the study and a waiver of the need for informed consent were obtained from the institutional review board of Sidra Medicine. All methods were carried out in accordance with relevant guidelines and regulations.

**Gene expression analysis.** Nucleic acids from NPS specimens were extracted on a Kingfisher Flex system using MagMAX viral/pathogen nucleic acid isolation kits (Thermo Fisher Scientific) according to the manufacturer's instructions. Predesigned TaqMan assays obtained from Thermo Fisher Scientific for two targets for ACE2 (Hs01085333_m1and Hs00222343_m1) and one target each for TMPRSS2 (Hs01122322_m1) and $\beta$-actin (Hs01060665) and a laboratory-developed assay for human RNase P were first assessed using a set of 10 random NPS specimens (5 positive and 5 negative for SARS-CoV-2) (data not shown). The expression of human RNase P in NPS specimens was assessed as described previously (21). The TaqMan assay for one of the ACE2 targets (Hs01085333_m1) was more sensitive (mean $C_T$ of 32.8 $\pm$ 2.6 versus 34.7 $\pm$ 2.2; $P = 0.00022$ by paired $t$ test) than that for the other and thus was chosen for the study. $\beta$-Actin was chosen as the endogenous control for the study because the analytical sensitivity of the assay was higher than that for RNase P, although the difference was not statistically significant (mean $C_T$ of 22.6 $\pm$ 2.6 for $\beta$-actin versus 23.7 $\pm$ 2.8 for RNase P). To determine the transcript levels of ACE2, TMPRSS2, and human $\beta$-actin (*ACTB*) genes in the study samples, 10 $\mu$l of the extracted nucleic acids were tested by the respective TaqMan assays using TaqPath 1-Step RT-qPCR master mix (Thermo Fisher Scientific), according to the instructions provided by the manufacturer. The transcript levels of the housekeeping gene $\beta$-actin were used to account for variation in the number of cells present in the NPS samples; the raw $C_T$ values for ACE2 and TMPRSS2 were normalized to the $C_T$ values for $\beta$-actin, and $C_T$ values relative to $\beta$-actin ($\Delta C_T$) were used as a measure of gene expression for all comparisons.

**Statistical analysis.** Descriptive statistics were used to summarize data on the study population and their SARS-CoV-2 infection status. Normality and log-normality of gene expression data were tested by the Shapiro-Wilk test and the Anderson-Darling test. The statistical significance of differences in ACE2 and TMPRSS2 expression in different groups was determined using the Kruskal-Wallis test with Dunn's multiple-comparison test and Mann-Whitney $U$ test for multiple groups and two groups, respectively. The association of ACE2 or TMPSS2 gene expression with SARS-CoV-2 RT-qPCR results (as the dependent variable) was tested in a multiple logistic regression model adjusted for age and sex for the overall population and adjusted for sex only for the subpopulation of patients from families with at least one adult family member positive for COVID-19. The association of age and sex with the expression of ACE2 and TMPRSS2 was tested in a multiple linear regression model with the transcript levels of the genes as dependent variables. The areas under the receiver operating characteristic curve in logistic regression analysis ranged from 0.57 to 0.69, and the $P$ values ranged from 0.04 to 0.0002. All analyses were carried out in GraphPad Prism version 9.0.2 for macOS (GraphPad Software, San Diego, CA).

## SUPPLEMENTAL MATERIAL

Supplemental material is available online only.

**SUPPLEMENTAL FILE 1**, PDF file, 0.1 MB.

## ACKNOWLEDGMENT

We are grateful to the Pathology Sciences laboratory of Sidra Medicine for providing specimens and laboratory facilities for the study.

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
