## [Reviewer comments · Microbiology Spectrum]

Microbiology Spectrum

Nasopharyngeal Expression of Angiotensin-Converting Enzyme-2 and Transmembrane Serine Protease-2 in Children Within SARS-CoV-2 Infected Family Clusters

Mohammad Hasan, Muneera Ahmad, Soha Dargham, Hatem Zayed, Alaa Al Hashemi, Nonhlanhla Ngwabi, Andres Perez-Lopez, Simon Dobson, Laith Abu Raddad, and Patrick Tang

Corresponding Author(s): Mohammad Hasan, Sidra Medicine

Review Timeline:

Submission Date:	July 6, 2021
Editorial Decision:	September 26, 2021
Revision Received:	October 3, 2021
Accepted:	October 9, 2021

Editor: Clinton Jones

Reviewer(s): The reviewers have opted to remain anonymous.

Transaction Report:

DOI: <https://doi.org/10.1128/Spectrum.00783-21>

September 26, 2021

Dr. Mohammad Rubayet Hasan
Sidra Medicine
Department of Pathology
Office no: H2M-24093
PO BOX 26999
Doha
Qatar

Re: Spectrum00783-21 (Nasopharyngeal Expression of Angiotensin-Converting Enzyme-2 and Transmembrane Serine Protease-2 in Children Within COVID-19 Family Clusters)

Dear Dr. Mohammad Rubayet Hasan:

Thank you for submitting your manuscript to Microbiology Spectrum. When submitting the revised version of your paper, please provide (1) point-by-point responses to the issues raised by the reviewers as file type "Response to Reviewers," not in your cover letter, and (2) a PDF file that indicates the changes from the original submission (by highlighting or underlining the changes) as file type "Marked Up Manuscript - For Review Only". Please use this link to submit your revised manuscript - we strongly recommend that you submit your paper within the next 60 days or reach out to me. Detailed information on submitting your revised paper are below.

Link Not Available

Sincerely,

Clinton Jones

Journals Department
Reviewer comments:

Reviewer #1 (Comments for the Author):

The paper write by Hasan and colleagues is well-constructed. It will be valuable to understand why children are more susceptible to COVID-19 infection.
Some elements need to be corrected.

Details:

The first sentence in the abstract section is little bit confusing. "accumulating evidences...may be related"

If we have all evidences? Rephrase it please

In line 38 "the odds of Covid-19 positivity increased" authors mean in children group? Please clarify this sentence.

Please for the reference 6 and its related sentence in the line 63 authors could add other references.

In line 74 theories or hypotheses?

For the sentence line 74-76 please add references about your assertions.
Line 106 "saved?" Or stored?
Replace Krustal wallis by "Kruskal-wallis" and after occurrences in the text.
Line 141 for Graphpad Prism 9 please add country and references.
Line 157 are or were triaged?
Lines 167-170 those lines are material and method section.
Some elements lines 177-179 could be move in the material and method section.
Same for line 188.
Line 193 mean or median Ct values?
Line 198 (W=value?)

Line 233 odds or odd ratio? (same in the abstract section?)
Lines 247-249 authors could add references.

Please for all significant tests mentioned in the text notice them by ** if their significant and also on your figures. If I well understand in the figure 2 none of tests are significant.
Moreover, in the figure 2 A) and B) the ANOVA hyphens over the figure are little bit shift.

In the supplementary figure 1 is not mean but median with IQR no? please verify

Supplementary figure 2: in the legend after the Shapiro-wilk test please check W value? (W=0.9822 ?)

Reviewer #2 (Comments for the Author):

Hasan and colleagues report that the ACE2 mRNA expression in NPS (as a surrogate for ACE2 expression in upper respiratory tract epithelium) was lower in children who were SARS-CoV-2 negative in exposed family (with at least one adult family member who were SARS-CoV-2 positive). Their family cohort design involving 105 families with 129 adults and 105 children, though using convenient leftover samples collected for precautionary testing from visitors to their medical unit, adds to the weight of ACE2 in SARS-CoV-2 pathogenesis. It should be noted that all subjects are asymptomatic or pre-symptomatic at the time of sample collection and the laboratory status of these initially SARS-CoV-2 negative cases could not be prospectively evaluated and confirmed.

The manuscript is easy to follow and data is well-presented. It could be improved by attention to the following points -

1. As the authors attempted to use the family cohort with SARS-CoV-2 exposure to illustrate that lower ACE2 expression in children was associated with lower chance of infection, the viral load of the infected adult family members was clearly one important confounding factor to the risk of virus transmission in household that was unfortunately not evaluated in the current study. It would be very helpful to include viral load data in their multivariate analysis if such information is available. This could substantially enhance their claim that SARS-CoV-2 infection status is ACE2 level dependent.
2. ACE2 mRNA level could not be determined from 58% (135/234) of samples. How many of them were ACE2 negative but SARS-CoV-2 positive? This large proportion appears to be problematic. Suggest adding a subgroup analysis of using samples with detectable ACE2 only to confirm their major observations.
3. The study was conducted between June 2020 and December 2020. It is presumed that all subjects were unvaccinated, and results are thus not confounded by vaccination status. This point should be mentioned in the Discussion.
4. Table 1: Indicate the unit of age.
5. Line 161 states that "the children and adults included in this study are asymptomatic at the time of NPS collection.". Therefore, it is a bit confusing that the term COVID-19 was used throughout the study (e.g., title, main text, tables and figures). Would it be more appropriate to use the term "SARS-CoV-2 infection" instead?

Staff Comments:

Preparing Revision Guidelines

To submit your modified manuscript, log onto the eJP submission site at <https://spectrum.msubmit.net/cgi-bin/main.plex>. Go to

Author Tasks and click the appropriate manuscript title to begin the revision process. The information that you entered when you first submitted the paper will be displayed. Please update the information as necessary. Here are a few examples of required updates that authors must address:

Please return the manuscript within 60 days; if you cannot complete the modification within this time period, please contact me. If you do not wish to modify the manuscript and prefer to submit it to another journal, please notify me of your decision immediately so that the manuscript may be formally withdrawn from consideration by Microbiology Spectrum.

October 9, 2021

Dr. Mohammad Rubayet Hasan
Sidra Medicine
Department of Pathology
Office no: H2M-24093
PO BOX 26999
Doha
Qatar

Re: Spectrum00783-21R1 (Nasopharyngeal Expression of Angiotensin-Converting Enzyme-2 and Transmembrane Serine Protease-2 in Children Within SARS-CoV-2 Infected Family Clusters)

Dear Dr. Mohammad Rubayet Hasan:

Both reviewers feel that you have addressed the concerns raised. Consequently, your manuscript has been accepted, and I am forwarding it to the ASM Journals Department for publication. You will be notified when your proofs are ready to be viewed.

Sincerely,

Clinton Jones
Editor, Microbiology Spectrum
